# Effects of Classroom Design on the Memory of University Students: From a Gender Perspective

**DOI:** 10.3390/ijerph18179391

**Published:** 2021-09-06

**Authors:** María Luisa Nolé, Juan Luis Higuera-Trujillo, Carmen Llinares

**Affiliations:** Institute for Research and Innovation in Bioengineering (i3B), Universitat Politècnica de València, 46022 Valencia, Spain; jlhiguera@i3b.upv.es (J.L.H.-T.); cllinare@omp.upv.es (C.L.)

**Keywords:** classroom design, learning processes, memory, gender, psychological responses, virtual classroom

## Abstract

Classroom design has important effects on the cognitive functions of students. However, this relationship has rarely been analysed in terms of gender. The aim of the present study, therefore, is to analyse the influence of different design variables (classroom geometry, wall colour, and artificial lighting) on university students’ memories from a gender perspective. To do so, 100 university students performed a memory task while visualising different design configurations using a virtual reality setup. Key results show that certain parameters, such as 5.23 m classroom width, 10,500 Kelvin lighting colour temperature, or the blue hue on the walls influence men and women in a similar way, while a purple hue or walls with low colour saturation can generate significantly different behaviour, especially in cognitive processes such as short-term memory. In this study, the use of virtual reality proved to be a useful tool to explore the design effects of virtual learning environments, increasingly present due to training trends and catalysed by the 2020 pandemic. This is a turning point and an international novelty as it will enable the design of classrooms (both physical and virtual) that maximise the cognitive functions of learners, regardless of gender.

## 1. Introduction

Most academic research in education has been focused on studying how to foster the internal aspects of students to improve their learning levels. Specific actions such as including family participation in the classroom [1], or the use of inverted methodology [2]—where students must study the content of the subject before class—promote learning to read and improve school performance, respectively. However, the reality is that learning is a very complex process influenced by many factors [3,4].

In this sense, there are far fewer studies that analyse classroom-contextual influences on the learning process. Although it is known that the configuration of the classroom, such as its layout or other elements of its design, relates to different cognitive [5] and emotional processes [6,7], few efforts have emerged to address these issues with respect to the existence of studies focused on understanding the influence of internal subject factors or teaching methods within a classroom.

This is of particular interest today as the 2020 pandemic steeply raises the importance of virtual learning (VR) environments. In recent months, the configurations of many classrooms have been greatly altered. The need for bubble groups in schools has forced a complete change in the distribution of furniture, and students have had to change their location with respect to other classmates, making use of new spaces that were initially created for other purposes, such as libraries or sports halls. Thus, it seems necessary to look more closely at the relationship between the physical learning environment and the students’ response to it.

Of all the design elements that make up the classroom, the environmental variables (temperature, acoustics, and lighting) have been the most studied. Studies such as that of Choi, Van Merriënboer and Paas [5] found that temperature and air quality were the most important contextual determinants of learning. In terms of acoustics, it has been found that the farther from the sound source and the greater the presence of noise, the greater the negative impact on the learning process [8,9]. Lighting has also been extensively studied for its involvement in physiological processes at the neurotransmitter level [10] and biological processes such as the regulation of circadian rhythms in humans [11]. For example, the presence of natural light has been identified as positively influencing reading and science activities [12,13]. In addition, the level of lighting affects cognitive performance depending on the difficulty of the task presented [14].

Another well-studied visual variable has been colour, because of its clear impact on students’ emotions and functionality [5,15]. Authors such as Nancy Kwallek have shown that more errors are made in reading and writing tasks in white spaces compared to coloured spaces [16]. However, there seem to be inconsistencies as to which colours generate better performance. Some authors relate differences to the age of the students [15,17,18], others point to the importance of task content [19], whilst a third group highlights the role of arousal in this interaction [20,21].

Finally, it is important to refer to the dimensions and geometry of a classroom, which is undoubtedly the least studied aspect [22]. The lack of studies that concretely analyse this variable makes it difficult to take a clear position on its importance. However, among those that have been carried out, it has been shown that classrooms with high ceilings have an impact on learning as they negate the benefit of better lighting as well as increasing acoustic problems due to reverberation [23]. 

In all cases, it is noteworthy that each of the spatial elements is approached in isolation, offering results that are not very decisive, as they do not take into account the integrated set of variables that contribute to the characteristics of the space. For this reason, studies are needed to analyse how several design variables can affect the cognition of the individual, making it possible to detect those that have a greater effect on their behaviour. Some studies have been based on contextualised points of interest through tasks of an attentional nature, with little consideration given to working memory, which is very much involved in the process of cognitive performance. Moreover, it is also important to note that research in this area does not pay much attention to the characteristics of the subject. The age of the subject is crucial given that learning is a continuous developmental process that varies over time [24]. Most studies focus on basic education at an early age, neglecting developments that may happen in higher education at the university level.

Yet another variable to be studied that is specific to the student is gender. Men and women show many differences in environmental perception in terms of colour preference [25] or sensitivity to glare in classroom lighting [26]. Igor Knez [27] found that long-term memory performance differed with respect to gender depending on the type of ambient lighting. Additionally, Hartstein [28] found an improvement in executive functions occurring with a higher level of lighting colour temperature. Physically, it is the temperature at which a black body should be heated to emit a certain colour of light. The lighting colour temperature is usually expressed in kelvins (K). At low temperatures the body would emit a light close to red (warm), and as the temperature increases it would be white (neutral) and later blue (cool). Although these gender differences are known, there has been little analysis of the interior of a classroom design from this perspective. 

From a methodological point of view, there is an important limitation in the reviewed studies regarding the stimuli used, as they generally employ physical spaces [17,29,30]. Physical spaces have the problem of fixing one aspect of the design. An extensive review of the application of virtual reality in research of sensory and perceptual science suggests VR as a good alternative [31]. There are studies that validate its use to generate and analyse different configurations of colour [32] and geometry [33]. These control an adequate level of sensation of presence for the subject, known as the “perceptual illusion of non-mediation” [34]. There are already numerous other studies that use VR to develop cognitive tasks that analyse aspects of attentional processes [35,36,37], or to train children with attention-deficit/hyperactivity disorder [38]. 

Based on the aspects detailed so far, the objective of this work is to analyse the impact that certain elements of classroom design—specifically room geometry, wall colour, and interior lighting—have on the memory of university students from a gender perspective. Memory is an important factor for academic performance. It is involved in the encoding and decoding of information. There is evidence that contextual factors such as lighting can especially affect short-term memory in virtual spaces [39], but also findings of a relationship between these variables in physical spaces [26,40,41]. Therefore, other factors could also be influencing this type of memory. Knowing this relationship would allow us to identify the design elements that positively or negatively affect the memory levels of female and male students, and inform the design of a classroom that benefits both genders.

## 2. Materials and Methods

To address the objective of the study, a laboratory experiment was conducted. It consisted of exposing participants to different experiences of a virtual classroom, modified in a controlled manner consisting of three variables: the geometry of the room; the colour of the walls, and artificial lighting. This was validated by quantifying the level of presence [42] of the virtual experiences (Phase I). In each virtual classroom, memory performance was measured for the female group (Phase II) and the male group (Phase III). Finally, the results of both groups were compared (Phase IV). Figure 1 shows the general methodological scheme.

### 2.1. Sample

A total of 100 participants were recruited for the experiment (mean age = 23.24 years; σ = 3.79). Care was taken to ensure that the sample was gender balanced: 50% male and 50% female. The primary inclusion criterion was to be a university student. In addition, two further conditions were established: (1) having Spanish nationality (to control for cultural variation); and (2) having normal or corrected-to-normal vision through contact lenses (avoiding spectacles given problems with usage of the VR device). Prior to the start of the experiment, all participants were duly informed of its workings after which they signed a consent form. The study was supervised by the Ethics Review Committee of the Institute for Research and Innovation in Bioengineering, at the Polytechnic University of Valencia, in compliance with the Declaration of Helsinki.

### 2.2. Stimuli

A classroom of the School of Building Engineering of the Polytechnic University of Valencia was chosen to be virtualised. The selection criterion was based on it being typical of the physical teaching spaces of the university. The virtual replica became the base classroom for subsequent variation. Modifications concerning room geometry, wall colour, and artificial lighting were then applied to it through the compensated change of the values of their respective parameters.

In the geometry variable, the parameters of height and width were considered. Specifically, six different measures of width and four of height were applied. These measures were based on a stepped increase or decrease in the original value by 1.2 m. The combination allowed for 24 geometrical modifications of the classroom (4 × 6, including the measurements of the base classroom). The length of the classroom was not modified.

For the colour variable, the parameters of hue and saturation were considered. The selection control was carried out using the Munsell notation system. Specifically, five hues equally distributed on the Itten colour circle [43], for each of which two saturations (high and low) were taken, separated by six Munsell chroma units. In combination it was then possible to obtain 10 modifications of the classroom (5 × 2, not including the measurements of the base classroom, because its colour was desaturated).

As for the lighting, both illuminance and colour temperature parameters were considered. Specifically, three luminance levels (being lower and higher than the value of the base classroom) and three colour temperature values (corresponding to values typically found on the market). The combination of these allowed 12 classroom modifications to be made (including the measurements of the base classroom).

It should be noted that each modification was applied separately to the base classroom, so the original parameters of the other two variables were kept constant. The combinations resulted in an array of classroom modifications that was administered to the participants. Each participant viewed four classrooms: the base, and three randomly modified virtual ones. Each virtual classroom was viewed from eight to nine times in total. Figure 2, Figure 3, Figure 4 describe the characteristics of each virtual classroom.

### 2.3. Scenario

Participants experienced the virtual classrooms through VR simulations on head-mounted displays (HMDs).

The VR classroom simulations were generated through a process of 3D modelling and rendering. For this, two main software packages were used: Rhinoceros (v.5.0, Robert McNeel & Associates, Seattle, WA, USA), and the Corona Renderer engine (v.2.0, Corona Renderer, Czech Republic, European Union). The colours of the walls studied were determined in the Munsell notation system, so for the virtual representation it was necessary to use ColorMunki^TM^ (X-Rite, Grand Rapids, MI, USA) to translate the five colours studied into RGB notation.

A head-mounted display (HTC Vive device) connected to the experimenter’s computer was used to simulate the classroom. Unity3D (v5.6, www.unity3d.com, accessed on 1 December 2004) was used to generate the software, which allowed the experimenter to show the different scenarios that the participant could then see through the HTC Vive. All the individual visualisations of the scenarios presented were identical. The subjects began the simulation, all seated at the same point in the classroom in the centre of the second row of tables. The participants could not get up and move around the room, but they could change the visual trajectory by means of head movements that allowed them to inspect the space. The corresponding memory test required wired speakers, which were calibrated before each session to ensure sound consistency.

### 2.4. Experimental Protocol

All experimental sessions were strictly conducted according to a protocol, always in the same room of the laboratory. The experimenter and participant were sat in opposition, each with their corresponding table, chair, and screen. The participant wore the HMD calibrated to the relevant environment. The experimenter’s position consisted of the computer from which the different virtual reality scenarios were launched and the memory test lists played. Attempts were made to control extraneous variables by running all experimental sessions during the same time period on each day. The experimenter tried to keep the level of acoustic isolation, the room temperature, and the furniture arrangement in the room unchanged. In addition, the devices used were the same for the whole sample. Table 1 shows the different actions carried out during the experimental protocol. Here it can be seen that actions B.2 to B.5 are repeated four times: first for the base virtual classroom, and then for three modified virtual classrooms.

### 2.5. Measurement

Data were collected from each participant in the virtual classroom to quantify firstly a sense of presence, and secondly, memory performance.

Sense of presence is defined as the feeling of being immersed in the VR and it not being perceived as artificial. For its evaluation, participants completed the SUS questionnaire [39], based on six items rated on a Likert-type scale (1–7). A quantification of presence was performed for each visualisation, by means of a “SUS-Total” metric. This questionnaire has been widely used to study and quantify the sense of presence of participants experiencing an environmental simulation using the same technology [44]. For every participant, the results of the six items were then totalled in each virtual classroom (out of a maximum of 42 points: 6 items * 7 points), to quantify the sense of presence of each visualisation. This resulted in the “SUS-Total” metric.

The memory task consisted of an instruction to memorise a set of words. Specifically, lists of 15 audible words using Loquendo TTS 7 (www.loquendo.com, accessed on 1 December 2004), were played on the same personal computer at the same volume level. The words were related by a common concept, but that concept was excluded from the list. Immediately after listening, subjects had to recall their maximum number of words within a time limit of 30 s. This is similar to the Deese, Roediger, and McDermott (DRM) paradigm experiments [45]. The task was repeated three times for each presentation with an interval of 2000 ms, but the words in each list were different. The order of presentation was randomised among a total of 12 different types of lists (four virtual classroom displays by three attention tasks during each display). Visual tasks were not used because it might interfere with the aim of the study on the effects of spatial configurations of an interior on academic performance, as it also requires the visual pathway. After testing, for every participant the memory performance was quantified as the total number of words correctly recalled in each virtual classroom (out of a maximum of 45 words: 15 words * 3 lists) based on Alonso et al. [46]. This resulted in the “Memory-Correct Answers” metric.

### 2.6. Data Analysis

Table 2 shows the analysis performed and the expected results for each Phase. IBM SPSS software was used (v.17.0, IBM, Armonk, NY, USA). The analysis of the level of sense of presence (Phase 1) was performed by summing the average level of the six items that compose the SUS questionnaire. The analysis of the effect of the different configurations of geometry, colour, and lighting on the memory of female (Phase 2) and male (Phase 3) students was carried out using statistical comparison techniques. The response of the students was compared to different categories of the design variables. This made it possible to identify differences and, if they existed, which design configuration was associated with higher or lower performance. In the same way, statistical comparison techniques were also applied to analyse the effect of classroom design on students’ memory as a function of gender (Phase 4). Thus, the incidence of each design configuration on memory was compared between male and female students. This analysis made it possible to identify significant differences between both genders. For the selection of these comparative statistical techniques, the criterion for performance normality of memory was verified using the Kolmogorov–Smirnov (K-S) test. This test determined a normal distribution of data (significance level > 0.05), so the statistical technique of analysis of variance (ANOVA) was applied. When the configurations to be compared had more than two categories, the Bonferroni post-hoc analysis was applied to identify the differences between them.

## 3. Results

The design effect was studied at the level of design parameter configuration (grouped with the other parameter of the same design variable), and not each combination independently. Therefore, 24 parameters of the variables geometry, colour, and lighting were studied (Table 3), which were combined to create the configurations shown in Figure 2, Figure 3, Figure 4. The statistical analysis of the data produced the following results.

### 3.1. Phase I. Analysis of Level of Sense of Presence

Mean levels of sense of presence per participant (based on the SUS questionnaire with the SUS-Total metric) were obtained for each environmental simulation. Taking into account the results obtained by studies using similar technologies [42], the presence levels were considered satisfactory. Figure 5 shows the average level of sense of presence for each simulated classroom.

### 3.2. Phase II. Effect of Classroom Design on the Memory of Female Students

In the following, the impact of design variations in shape, colour, and lighting on the performance of the memory task is analysed for female university students. In the psychological memory task, the Memory-Correct Answers metric was used. This quantifies the number of words recalled in the psychological memory task. Simply put, the more words recalled, the better the memory performance. Due to the normality of these data (K-S, *p* > 0.05), an ANOVA was applied. Figure 6 shows the normalised means for the set of configurations analysed.

#### 3.2.1. Effect of Variations in Classroom Shape (Ceiling Height and Classroom Width)

The ANOVA found no significant differences in memory task performance when changing the ceiling height (*p* < 0.05). 

However, they did occur when varying the width of the classroom (*p* = 0.002), with a significant decrease in memory-task performance when reducing the width of the classroom. Thus, a relevant change was observed with measurements of less than 7.2 m. The worst result in the memory test corresponded to a width of 3.6 m. A Bonferroni post-hoc analysis showed that this difference occurred between 3.6 m and the measurements of 7.2 m (*p* = 0.048).

#### 3.2.2. Effect of Variations in Classroom Colour (Hue and Saturation)

Classroom colour saturation was shown to significantly influence memory task performance for female students (*p* = 0.023). Accordingly, ANOVA showed significantly higher results for higher saturations.

Additionally, the hues of the classroom coverings generated significant differences in the memory test (*p* = 0.000). The best result was observed with the 5B (blue) hue and the worst performance with the 5Y (yellow) and 5P (purple) hues, demonstrating significant differences between them (5B vs. 5Y and 5B vs. 5P) (*p* = 0.000).

#### 3.2.3. Effect of Variations in Classroom Lighting (Colour Temperature and Illuminance)

The colour temperature of the lighting was also an element affecting memory, with significant differences depending on the colour temperature (*p* = 0.000). It was observed that the higher the colour temperature, the better the memory task performance. However, this relationship reached a turning point between 6500 and 10,500 K, where the memory test result then decreased significantly. Thus, the best result corresponded to the colour temperature of 6500 K, while the worst result was for 10,500 K. The Bonferroni post-hoc analysis showed significant differences between the lowest and highest colour temperatures (3000 vs. 6500 K, *p* = 0.001; 3000 vs. 10,500 K, *p* = 0.033; 4000 vs. 6500 K, *p* = 0.000; 4000 vs. 10,500 K, *p* = 0.012) and between the two highest colour temperatures, between which there was a change in trend (6500 K vs. 10,500 K, *p* = 0.000). Illuminance, however, did not influence the results of the memory test.

### 3.3. Phase III. Effect of Classroom Design on Male Students’ Memory

This section analyses the effect that design variations have on memory test performance for males. Again, we use the Memory-Correct Answers metric, which quantifies the number of words remembered in the psychological memory task. Due to the normality of these data (K-S, *p* > 0.05), an ANOVA was applied. Figure 7 shows the normalised means obtained for the set of configurations analysed.

#### 3.3.1. Effect of Variations in Classroom Shape (Ceiling Height and Classroom Width)

ANOVA showed that there was no significant difference in memory task performance when modifying ceiling height (*p* < 0.05). However, the width of the classroom did seem to be a determining factor in the memory test, as the ANOVA identified significant differences in the results obtained in the memory test when varying the width of the classroom (*p* = 0.024). The best result corresponded to the dimensions of 8.4 m, while the worst performance was obtained with a width of 2.4 m.

#### 3.3.2. Effect of Variations in Classroom Colour (Hue and Saturation)

Classroom colour saturation did not influence memory task performance for male students (*p* = 0.023). Higher mean values were found for low saturations versus high saturations. Nevertheless, hue variations did generate significant differences in the memory test (*p* = 0.000). The best results were observed with the 5P (purple) hue and the worst with the 5G (green) and 5Y (yellow) hues, with significant differences between them (5P vs. 5G and 5P vs. 5Y) (*p* = 0.000).

#### 3.3.3. Effect of Variations in Classroom Lighting (Colour Temperature and Illuminance)

The colour temperature of the lighting had a significant effect on memory test performance (*p* = 0.022). In this case, positive results were observed for lighting colour temperature values between 3000 and 6500 K, but there was a negative result when moving to 10,500 K. The Bonferroni post-hoc analysis showed an inflection point when going from 6500 to 10,500 K (*p* = 0.043), similarly in the case of women. The illuminance did not influence the results of the memory test for this group.

### 3.4. Phase IV. Comparative Analysis of the Effect of Classroom Design on Students’ Memory as a Function of Gender

Here, the differences in the analysed design configurations as a function of gender are analysed. Figure 8 shows the normalised means of both genders on the same graph.

#### 3.4.1. Significant Differences by Gender for Changes in the Shape of the Classroom (Ceiling Height and Classroom Width)

Although for either gender, the ceiling height did not influence performance on the memory task, different results were nevertheless observed. In general, the results of males were higher than those of females for the different ceiling heights analysed. The ANOVA found significant differences in memory task performance for the height of 3.2 m (*p* = 0.000).

Regarding the width of the classroom, it was observed that as the width of the classroom increased, better memory test results were produced in both groups. The change in trend in the results for men occurred from a width of 3.6m, whereas for women this change was observed from 7.2 m. It was this difference that produced significant differences for the width of 3.6 (*p* = 0.037).

#### 3.4.2. Significant Differences by Gender for Changes in Classroom Colour (Hue and Saturation)

Classroom colour saturation had a significant influence only for the women’s group. However, a significantly different result (*p* = 0.001) was observed for both genders in the case of stimuli with low colour saturation. Thus, low saturation appeared to generate positive results in men and very negative results in women.

With respect to hue, significantly different results were observed between males and females. The most pronounced difference was found in the 5P hue (purple) with a very positive result for men and a very negative result for women (*p* = 0.000). Significant differences were also detected in the 5Y (green) hue (*p* = 0.040).

#### 3.4.3. Significant Differences by Gender for Changes in Classroom Lighting (Colour Temperature and Illuminance)

The colour temperature of the lighting had a similar effect. Significant differences were detected at the extreme levels. In the case of the lowest colour temperature, 3000 K, memory test performance was above average for males and below average for females (*p* = 0.009). For the highest level of colour temperature, 10,500 K, a very negative performance was observed for both groups, but significantly lower for females (*p* = 0.000).

In the case of illuminance, differences were observed at the 300 lx value, with above average performance for males and below average for females (*p* = 0.042).

## 4. Discussion

How the external context of a subject influences internal cognitive processes is a paradigm widely studied in the field of environmental psychology. Scientific analysis of aspects of spatial design in this field seeks to link two areas of knowledge that are apparently quite distinct: architecture, and educational psychology. Accepting the influence of external stimuli does not make the teacher’s work less relevant, nor does it diminish examination of the students’ internal perceptions. Rather, it serves to encourage a favourable outcome through the control of the physical variables of the given environment.

At a methodological level, the use of VR for spatial simulation should be highlighted, as it allows some variables under study to be easily modified while controlling others. In recent years, the use of VR in academic fields has become particularly relevant, with the year 2020 being a turning point on a global level due to the public health situation. Clearly, there are countless reasons for the use of this new technology to be implemented in our societies, with specific VR design for activities to be developed from an evidential base. In this sense, studies that contribute to improving the design of these new learning environments are particularly relevant.

Regarding the results obtained, it is important to note that the cognitive performance of women in the memory test was generally lower than that of men. This phenomenon is possibly due to the task being carried out in a virtual environment. In this regard, Barrett and Lally [47] showed that men and women play different social roles in the use of learning technologies. It has also been found that men have a greater preference for the use of multimedia technologies for learning due to a greater perceived usefulness of electronic devices [48], familiarity with video games [49,50], and a greater use of VR [51]. There is also research conducted from virtual classrooms where differences are observed between men and women in terms of classroom design preferences [52].

Moreover, there is also evidence that learning differences between genders at school age are associated with task content [53]. Some authors conclude that gender difference could be correlated with the nature of spatial memory tasks [54,55,56,57] that relate to different forms of orientation and activation of brain structures [54,58,59]. In tasks that involve technological elements, men show greater mental rotation than women [55,56]. For this, men had greater activation of hippocampal structures while women of more cortical structures such as the frontal lobe [59]. This may imply different ways of perceiving the virtual environment that act upon, albeit indirectly, the employment of memory.

When carrying out tasks in virtual environments, it is necessary to explore whether the differences obtained depend in a decisive way on the technology itself, on previous experience with the devices, or on the type of task set out. The specific results of classroom design do show that there are some elements of classroom design that influence both men and women in the same way, and others that differ in their influence according to gender. Knowing these differences makes it easier to suggest the creation of gender-equitable virtual environments.

Specifically speaking, ceiling height and illuminance do not influence memory performance. However, a small classroom width (2.4 m), and a high lighting colour temperature (10,500 K and above) have a negative impact on the memory task in both groups. However, the colour variable shows a large gender-related difference. Thus, women seem to be more sensitive to colour saturation as has been found in non-virtual (physical) environments [26]. It has been found that that low saturation used in the classroom generates negative results for women. The 5P (purple) hue has been identified as beneficial for males, but not for females. Based on this, the combination 5B (blue) and high saturation would be a good alternative for both genders, as it offers the best memory results for both groups without significant differences between them.

Finally, although the use of VR is a good tool to carry out this study, it cannot be used by students with glasses. This is a considerable limitation of the study. On the other hand, in terms of the limitations, it is important to take into account the possibility of synergistic effects between design elements not explored in this study. The “ceteris paribus” experimental logic applied in this study (which involves modifying each of the variables individually, keeping all other variables constant) gives rise to two issues: (1) there could be significant effects when combining the variables studied (geometry, colour, and lighting); and (2) other design variables to-date not analysed (for example, the position and design of the furniture or the virtual position of the participant in the classroom), by remaining the same in all experiences, could limit them to relatively similar situations

## 5. Conclusions

This study analyses the influence of different physical aspects of a classroom on students’ memory from a gender perspective. The results show relevant differences in the effects that design has on memory depending on gender, especially with purple wall colour and low saturation. This provides relevant information for the design of classrooms that benefit both groups.

Lighting, colour, and classroom dimensions are fundamental aspects to consider in the design of architectural teaching spaces, but they could also be of interest for the design of virtual learning environments that are increasingly present in our society. The use of VR in this study allows us to explore this. In educational settings, it has been suggested that it may offer even a better experimental context than physical settings [35,60,61]. In addition, this has allowed us to do something that was difficult until now: to study the influence of architectural elements on human behavior, overcoming the difficulties and cost of a real construction.

In this sense, the results may be of interest to design professionals or architects of classrooms in order to maximise the cognitive functions of students, regardless of gender. Research can also contribute to students’ ability to shape space in virtual reality classrooms. However, it should not be forgotten that personal preferences of perception do not always coincide with a better level of memory. On the other hand, they may be relevant for teachers looking to adapt their teaching methodology given the existing physical characteristics of the classroom.

It would be interesting to analyse the joint effect of the combination of the studied variables and to see if these results are sustained in substantially different classroom typologies. Furthermore, future research may further address this question and whether gender differences are maintained over the years or whether, on the contrary, it is characteristic of youth and limited to learning contexts.

## Figures and Tables

**Figure 1 ijerph-18-09391-f001:**
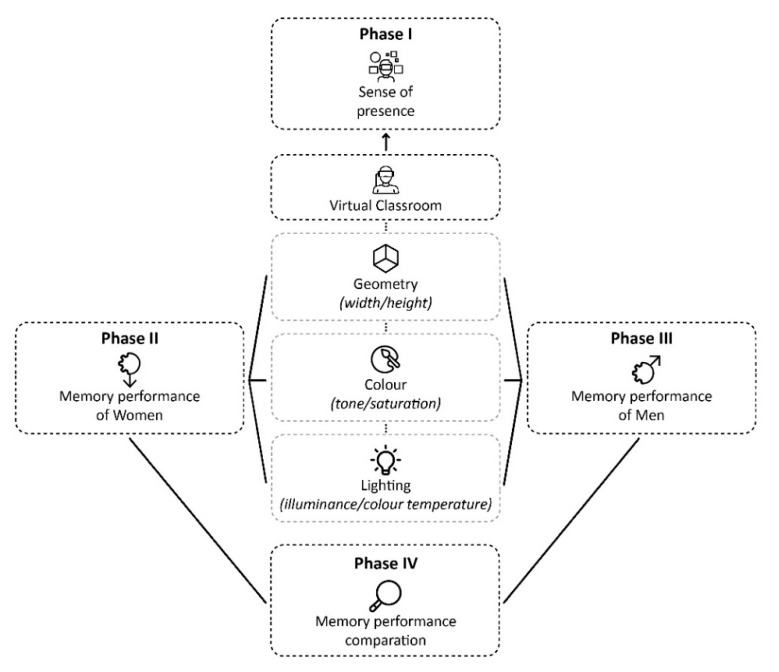
General methodological scheme.

**Figure 2 ijerph-18-09391-f002:**
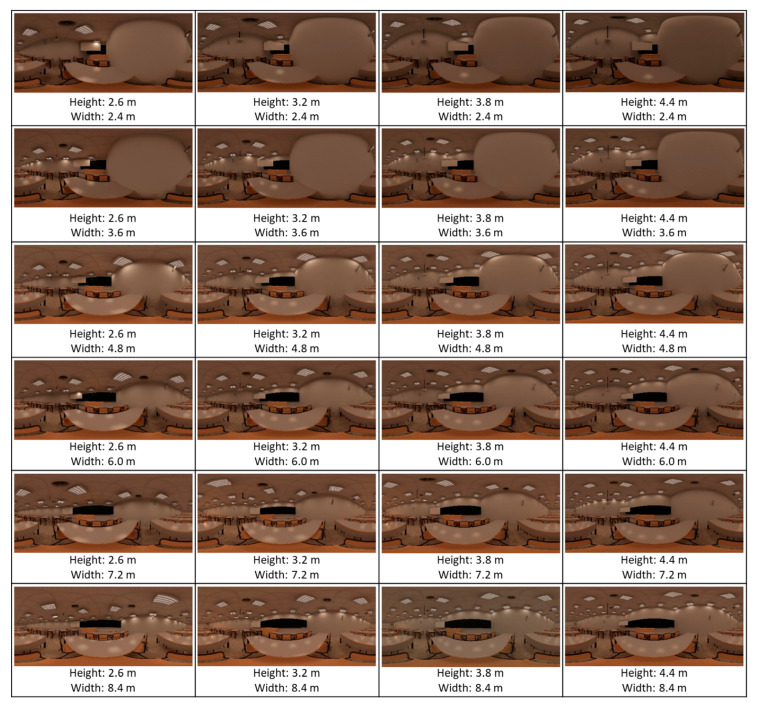
Virtual classrooms modified in geometry.

**Figure 3 ijerph-18-09391-f003:**
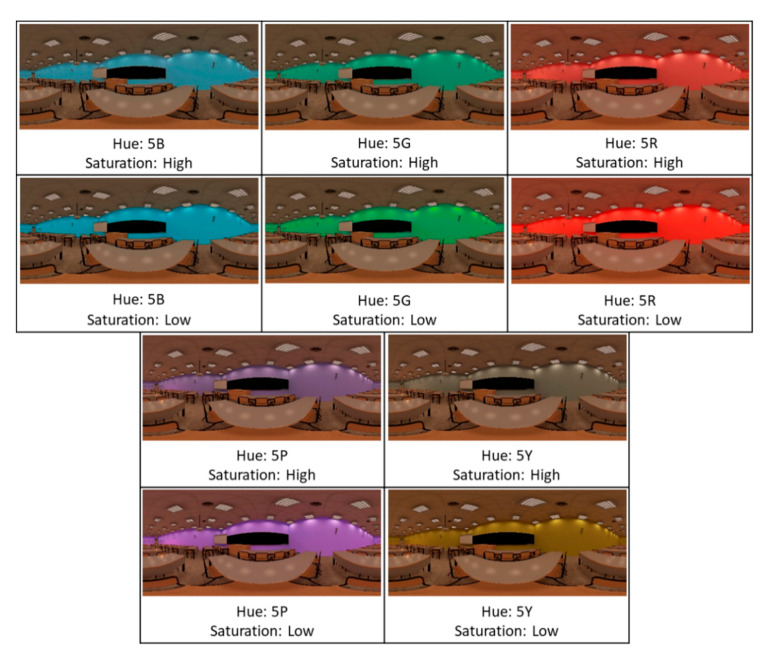
Virtual classrooms modified in colour.

**Figure 4 ijerph-18-09391-f004:**
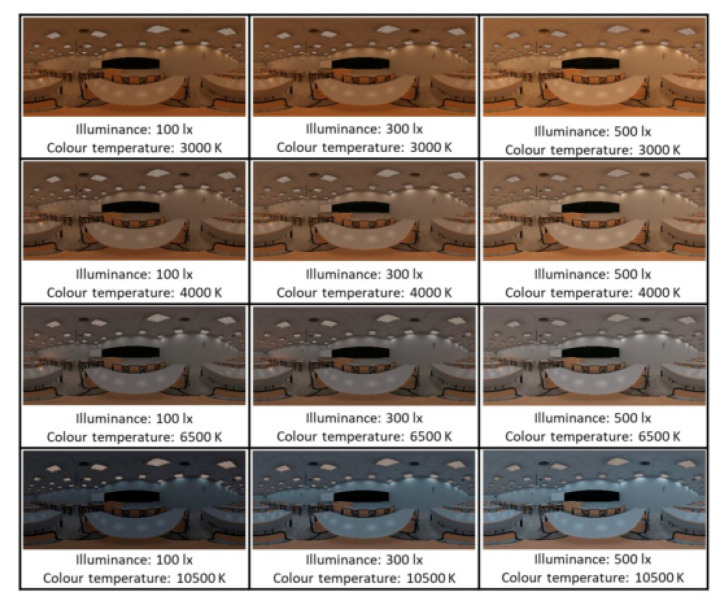
Virtual classrooms modified in lighting.

**Figure 5 ijerph-18-09391-f005:**
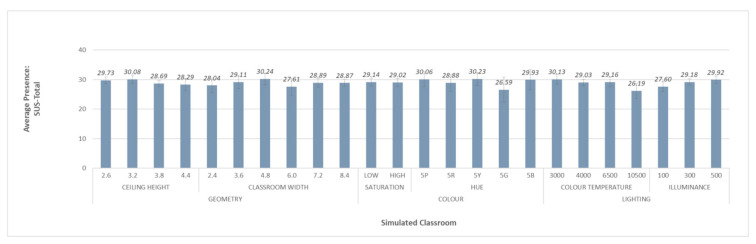
Average level of presence for each simulated classroom.

**Figure 6 ijerph-18-09391-f006:**
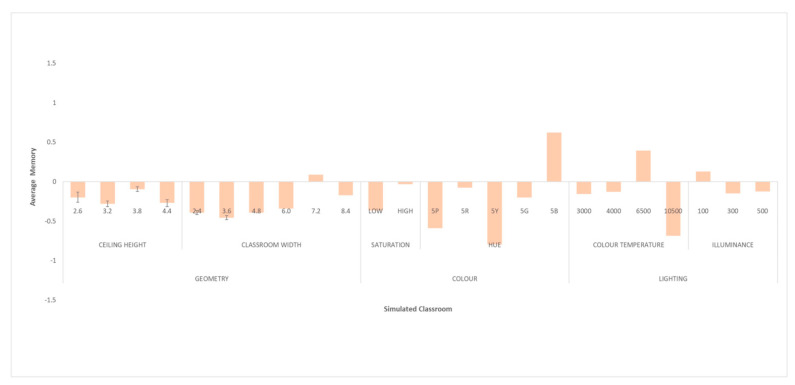
Standardised means of the psychological memory task for women, Memory-Correct Responses for women.

**Figure 7 ijerph-18-09391-f007:**
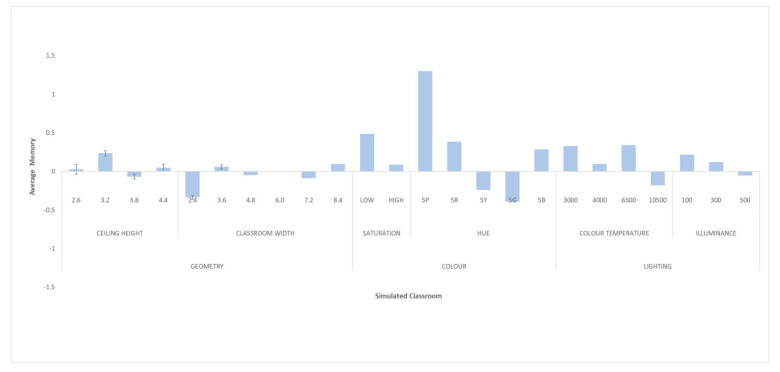
Normalised means of Memory-Correct Answers in men’s sample.

**Figure 8 ijerph-18-09391-f008:**
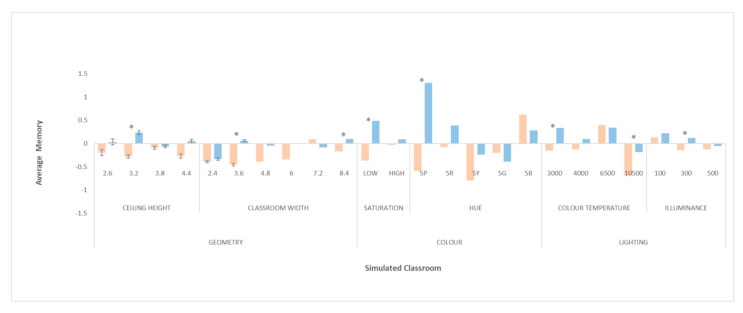
Normalised means of the psychological memory task. Memory-Correct Answers for men are shown in blue; and for women shown in orange. An asterisk marks the significant differences between genders.

**Table 1 ijerph-18-09391-t001:** Experimental protocol.

**A. Preparation**		**A.1. Welcome of the participant:** Reception of the participant, information about the procedure, signing of documents, and resolution of doubts.
		**A.2. Visual colour perception test:** Administration of the Farnsworth–Munsell Dichotomous D-15 test, to detect possible problems in colour vision.
		**A.3. VR test:** Placement of the virtual reality device, and presentation of a neutral scenario (different from the stimulus) to settle the participant’s habituation, and any further problem solving.
**B. Experimental**		**B.1. Initiation of the experimental phase:** Administration of instructions on the start of the experimental phase via the computer screen. After this, final positioning of the virtual reality device.
	**For each** **Virtual** **Classroom**	**B.2. Relaxing audio:** Administration of natural sounds for 60 s, to promote a state of relaxation and reduce the participant’s mental fatigue.
		**B.3. Virtual classroom:** Visualisation of the virtual classroom for 90 s.
		**B.4. Memory assessment:** Administration of the memory task, and quantification of the number of correctly recalled words (Memory-Correct Answers metric).
**C. Post-** **Experiment**		**C.1. End of the experimental phase:** Final removal of the virtual reality device, followed by administration of the instructions on the end of the experimental phase via the computer screen.
		**C.2. Demographic data:** Completion of a basic demographic questionnaire.
		**C.3. Participant’s farewell:** Informing of the end of the process, resolution of any doubts, and indication of the exit.

**Table 2 ijerph-18-09391-t002:** Statistical treatments.

Phase	Analysis	Statistical Treatment	Expected Result
Phase 1 Validation of the VR environment.	Analysis of level of sense of presence. SUS-Total.	Descriptive analysis of means.	Sufficient level of presence.
Phase 2 Effect of classroom design on the memory of female students.	Analysis of memory performance (women), by modifying the shape, colour and lighting of the classroom. Memory-Correct answers (women)	ANOVA and Bonferroni’s post-hoc analysis (normally distributed data) for memory hits, according to the different categories of the design variables.	Significant differences in memory performance, depending on the classroom design. Identification of the design which gave the best and the worst memory performance, for the women’s group.
Phase 3 Effect of classroom design on the memory of male students.	Analysis of memory performance (men), by modifying the shape, colour, and lighting of the classroom. Memory-Correct answers (men)	ANOVA and Bonferroni’s post-hoc analysis (normally distributed data) for memory hits, according to the different categories of the design variables.	Significant differences in memory performance, depending on the classroom design. Identification of the design which gave the best and the worst memory performance, for the men’s group.
Phase 4 Comparative analysis of the effect of classroom design on students’ memory as a function of gender.	Comparative analysis of memory performance for each configuration by gender.	ANOVA and Bonferroni’s post-hoc analysis (normally distributed data) for memory hits, according to gender.	Significant differences in memory performance, relating to their gender

**Table 3 ijerph-18-09391-t003:** Set of parameters studied.

Geometry		Colour		Lighting	
Ceiling Height	Classroom Width	Saturation	Hue	Colour Temperaure	Illuminance
2.6 m	2.4 m	Low	5P	3000 K	100 lx
3.2 m	3.6 m	High	5R	4000 K	300 lx
3.8 m	4.8 m		5Y	6500 K	500 lx
4.4 m	6 m		5G	10,500 K	
	7.2 m		5B		
	8.4 m				

## Data Availability

The data presented in this study are available on request from the corresponding author.

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
