# Peer review of "Effects of Classroom Design on the Memory of University Students: From a Gender Perspective"

_ijerph, 2021, doi:10.3390/ijerph18179391_

Round 1
Reviewer 1 Report
It is an interesting topic: Effects of Classroom Design on the Memory of University Students: From a Gender Perspective. However, the paper needs to clarify some key information to help with the readability.
1)In abstract, summarized the key results and research novelty should be stressed, which is missing in the current manuscript.
2)The aim of the paper is to analyse the influence of different design variables (classroom geometry, wall colour, and 10 artificial lighting) on university students’ memory from a gender perspective (Line 9-11, Page 1). Hence, Memory is at the heart of this article, and authors needs to clarify the rationale in last paragraph, Section 1 and the 1st paragraph, Section 2. For example, what is the rationale and theory behind the colour, lighting and geometry have impact on what kind of memory (Line 95-97, Page 2) and memory performance (Figure 1), which needs more literature to support.
3)For research methods, in terms of research aim and objectives, authors needs to clarify why and what is the adopted methods, such as SUS questionnaire (Line 195, Page 7), the SUS questionnaire with the SUS-Total metric (Line 222, Page 9),‘Memory-Correct Answers’ metric (Line 210, Page 8) , and ANOVA (in Table 2; Line 235, Page 9).
4)For research results, in terms of research aim and objectives, authors needs to clarify what are “24 parameter configurations were studied” (Line 219, Page 9).
5)In discussion, the research limitation (Line 384, Page 13) needs to further clarify why the “the possibility of synergistic effects between design elements not explored in this study” (Line 384-385, Page 13), and “Although each of the variables has been modified while holding others constant, there may still be unseen effects between other design variables to-date not analysed ” (Line 385-387, Page 13), which is the key aspects to the research aim. Besides, what are “each of the variable”(Line 385-386, Page 13)? The limitation of research environment, i.e. Virtual Reality environment V.S. real physical environment should be discussed.
6)The conclusion section needs to summarize the key results, research novelty, and future research, which is missing in the current manuscript.
Author Response
1. It is an interesting topic: Effects of Classroom Design on the Memory of University Students: From a Gender Perspective. However, the paper needs to clarify some key information to help with the readability.
We thank the reviewer for his interest in the topic of this article. Each of his recommendations serves to improve the understanding of the article.
2. In abstract, summarized the key results and research novelty should be stressed, which is missing in the current manuscript.
In fact the abstract is an important part of the text. We agree with the reviewer that the importance of the results and the novelty of the research are not emphasised enough. In this sense, we have followed the recommendations to highlight the most important results and the novelty of the research. In order to comply with the publisher's word limit, we have made brief modifications. The final summary is as follows:
"Classroom design has important effects on the cognitive functions of students. However, this relationship has rarely been analysed in terms of gender. The aim of the present study, therefore, is to analyse the influence of different design variables (classroom geometry, wall colour, and artificial lighting) on university students’ memory from a gender perspective. To do so, 100 university students performed a memory task while visualising different design configurations using a virtual reality setup. Key results show that certain parameters, such as 5.23 metre classroom width, 10,500 Kelvin lighting colour temperature or the blue hue on the walls, influence men and women in a similar way, while a purple hue or walls with low colour saturation can generate significantly different behaviour especially in cognitive processes such as short-term memory. In this study, the use of virtual reality proved to be a useful tool to explore the design effects of virtual learning environments, increasingly present due to training trends and catalysed by the 2020 pandemic. This is a turning point and an international novelty as it will enable the design of classrooms (both physical and virtual) that maximise the cognitive functions of learners, regardless of gender."
3. The aim of the paper is to analyse the influence of different design variables (classroom geometry, wall colour, and 10 artificial lighting) on university students’ memory from a gender perspective (Line 9-11, Page 1). Hence, Memory is at the heart of this article, and authors needs to clarify the rationale in last paragraph, Section 1 and the 1st paragraph, Section 2. For example, what is the rationale and theory behind the colour, lighting and geometry have impact on what kind of memory (Line 95-97, Page 2) and memory performance (Figure 1), which needs more literature to support.
According to the reviewer's comments, it is necessary to expand the bibliography of how certain physical aspects such as lighting affect specifically short-term memory. Bearing this in mind, we asked, could other parameters such as color and geometry also be influencing this type of memory? Therefore, we have reflected this in the following paragraph:
“Based on the aspects detailed so far, the objective of this work is to analyse the impact that certain elements of classroom design—specifically: interior lighting; room geometry; and wall colour—have on the memory of university students from a gender perspective. Memory is an important factor for academic performance. It is involved in the encoding and decoding of information. There is evidence that contextual factors such as lighting can especially affect short-term memory in virtual spaces [39], but also finding a relationship between these variables in physical spaces [26,40,41]. So other factors could also be influencing this type of memory. Knowing this relationship would allow us to identify the design elements that positively or negatively affect the memory levels of female and male students, and inform the design of a classroom that benefits both genders.”
4. For research methods, in terms of research aim and objectives, authors needs to clarify why and what is the adopted methods, such as SUS questionnaire (Line 195, Page 7), the SUS questionnaire with the SUS-Total metric (Line 222, Page 9), ‘Memory-Correct Answers’ metric (Line 210, Page 8) , and ANOVA (in Table 2; Line 235, Page 9).
The underlying rationale for using the SUS questionnaire, the memory task (quantified by the metrics described) and ANOVA has been clarified. As the reviewer suggests, the paper has benefited from including this information.
“Sense of presence is defined as the feeling of being immersed in the VR and not being perceived as artificial. For its evaluation, participants completed the SUS questionnaire [39], based on six items rated on a Likert-type scale (1-7). This questionnaire has been widely used to study and quantify the sense of presence of participants experiencing an environmental simulation using the same technology. [44]. For every participant, the results of the six items were then totalled in each virtual classroom (out of a maximum of 42 points: 6 items * 7 points), to quantify the sense of presence of each visualisation. This resulted in the 'SUS-Total' metric.”
“The memory task consisted of an instruction to memorise a set of words. Specifically, lists of 15 audible words using Loquendo TTS 7 (www.loquendo.com), were played on the same personal computer at the same volume level. The words were related by a common concept, but that concept was excluded from the list. Immediately after listening, subjects had to recall their maximum number of words within a time limit of 30 seconds. This is similar to the Deese, Roediger, and McDermott (DRM) paradigm experiments [41]. The task was repeated three times for each presentation with an interval of 2,000 milliseconds, but the words in each list were different. The order of presentation was randomised among a total of 12 different types of lists (four virtual classroom displays by three attention tasks during each display). Visual tasks were not used because it might interfere with the aim of the study on the effects of spatial configurations of an interior on academic performance, as it also requires the visual pathway. After testing, for every participant the memory performance was quantified as the total number of words correctly recalled in each virtual classroom (out of a maximum of 45 words: 15 words * 3 lists) based on Alonso et al [42]. This resulted in the ‘Memory-Correct Answers’ metric”.
“Table 2 shows the analysis performed and the expected results for each Phase. IBM SPSS software was used (v.17.0, www.ibm.com/products/spss-statistics). The analysis of the level of sense of presence (Phase 1) was performed by summing the average level of the six items that compose the SUS questionnaire. The analysis of the effect of the different configurations of lighting, geometry and colour on the memory of female (Phase 2) and male (Phase 3) students was carried out using statistical comparison techniques. It was compared the response of the students to different categories of the design variables. This made it possible to identify differences and, if they existed, which design configuration was associated with higher or lower performance. In the same way, statistical comparison techniques were also applied to analyse the effect of classroom design on students' memory as a function of gender (Phase 4). Thus, the incidence of each design configuration on memory was compared between male and female students. This analysis made it possible to identify significant differences between both genders. For the selection of these comparative statistical techniques, the criterion for performance normality of memory was verified using the Kolmogorov-Smirnov (K-S) test. This test determined a normal distribution of data (significance level >0.05), so the statistical technique of analysis of variance (ANOVA) was applied. When the configurations to be compared had more than two categories, the Bonferroni post-hoc analysis was applied to identify the differences between them.”
5. For research results, in terms of research aim and objectives, authors needs to clarify what “24 parameter configurations were studied” (Line 219, Page 9).
We appreciate your time to highlight this point. The 24 parameters referred to in the text are each of the studied values of each variable of colour, illumination and geometry. To clarify this, the following sentence has been used in concrete terms:
“Therefore, 24 parameters of the variables lighting, geometry and colour were studied (Table 3), which were combined to create the configurations shown in figures 2-4.”
For a better understanding of the text, Table 3 has been added, a summary of these 24 parameters, which correspond to the figures of the analyzes. Figure 2 has also been modified in a more organized way.
6. In discussion, the research limitation (Line 384, Page 13) needs to further clarify why the “the possibility of synergistic effects between design elements not explored in this study” (Line 384-385, Page 13), and “Although each of the variables has been modified while holding others constant, there may still be unseen effects between other design variables to-date not analysed ” (Line 385-387, Page 13), which is the key aspects to the research aim. Besides, what are “each of the variable”(Line 385-386, Page 13)? The limitation of research environment, i.e. Virtual Reality environment V.S. real physical environment should be discussed.
The limitations of the study have been better discussed. We thank the reviewer for his suggestion, as it contributes to a better quality of the paper.
“Finally, although the use of VR is a good tool to carry out this study, it cannot be used by students with glasses. This is a considerable limitation of the study. On the other hand, in terms of the limitations, it is important to take into account the possibility of synergistic effects between design elements not explored in this study. The "ceteris paribus" experimental logic applied in this study (which involves modifying each of the variables individually, keeping all other variables constant) gives rise to two issues: (1) there could be significant effects when combining the variables studied (geometry, colour, and lighting); and (2) other design variables to-date not analysed (for example, the position and design of the furniture or the virtual position of the participant in the classroom), by remaining the same in all experiences, could limit them to relatively similar situations”
7. The conclusion section needs to summarize the key results, research novelty, and future research, which is missing in the current manuscript.
It is important to note the reviewer's appreciation of the discussion, as this is a section that needs to be improved. According to the reviewer’s suggestions, the discussion has been extended:
“This study analyses the influence of different physical aspects of a classroom on students’ memory from a gender perspective. The results show relevant differences in the effects that design has on memory depending on gender, especially with purple wall colour and low saturation. This provides relevant information for the design of classrooms that benefit both groups.
Lighting, colour, and classroom dimensions are fundamental aspects to consider in the design of architectural teaching spaces, but they could also be of interest for the design of virtual learning environments that are increasingly present in our society. The use of VR in this study allows us to explore this. In educational settings, it has been suggested that it may offer even a better experimental context than physical settings [35, AA]. In addition, this has allowed us to do something that was difficult until now: to study the influence of architectural elements on human behavior, overcoming the difficulties and cost of a real construction.
In this sense, the results may be of interest to design professionals or architects of classrooms in order to maximise the cognitive functions of students, regardless of gender. Research can also contribute to students' ability to shape space in virtual reality classrooms. However, it should not be forgotten that personal preferences of perception do not always coincide with a better level of memory. On the other hand, they may be relevant for teachers looking to adapt their teaching methodology given the existing physical characteristics of the classroom.
It would be interesting to analyze the joint effect of the combination of the studied variables and to see if these results are sustained in substantially different classroom typologies. Furthermore, future research may further address this question and whether gender differences are maintained over the years or whether, on the contrary, it is characteristic of youth and limited to learning contexts”
Reviewer 2 Report
Introduction- please explain 'inverted methodology' to reader.
p.1 line 31. you say 'few efforts have been made' but then discuss plenty of studies- can you clarify in which area the lack of research exists please?
Explain lighting colour temperature please.
the exclusion of people wearing spectacles seems to be a limitation- many students will wear spectacles- address?
p.7 line 184- missing word 'time'.
Could the option of the student setting the colour, geometry of the room be a future possibility?
Author Response
We are grateful to the reviewer for his valuable comments. We appreciate his time and effort in improving the manuscript.
- Introduction- please explain 'inverted methodology' to the reader.
Inverted methodology is a concept that is widely used in the field of teaching.We agree with the reviewer that this term required further explanation. This allows to improve the reading of all the profiles of readers interested in the article. In accordance with the reviewer's suggestions, the paragraph on inverted methodology has been expanded as follows:
“[...]or the use of inverted methodology [2] — where students must study the content of the subject before class — promote learning to read and improve school performance, respectively.”
- p.1 line 31. you say 'few efforts have been made' but then discuss plenty of studies- can you clarify in which area the lack of research exists please?
This sentence may mislead the reader. It is intended to convey the idea that, although there is evidence of the influence of the environment on internal cognitive processes, there is little research that examines this in relation to the amount of research that addresses the influence of intra-subject or inter-subject factors on these cognitive processes. The paragraph has therefore been modified:
“few efforts have emerged to address these issues with respect to the existence of studies focused on understanding the influence of internal subject factors or teaching methods within a classroom.”
- Explain lighting colour temperature please.
Indeed, this term needed to be clarified. Following the reviewer's suggestion, its description has been added in the introduction, since it is the first time this term is mentioned.
“[...] Also, Hartstein [28] found an improvement in executive functions occurring with a higher level of lighting colour temperature. Physically, it is the temperature at which a black body should be heated to emit a certain colour of light. The lighting colour temperature is usually expressed in kelvins (K). At low temperatures the body would emit a light close to red (warm), and as the temperature increases it would be white (neutral) and later blue (cool). Although these gender differences are known, there has been little analysis of the interior of a classroom design from this perspective.”
- the exclusion of people wearing spectacles seems to be a limitation address?
We thank the reviewer for this suggestion. It is true that this may be a limitation for the research. Therefore, it has been added to the conclusion section:
“Finally, although the use of VR is a good tool to carry out this study, it cannot be used by students with glasses. This is a considerable limitation of the study. On the other hand, in terms of the limitations, it is important to take into account the possibility of synergistic effects between design elements not explored in this study [...]”.
- p.7 line 184- missing word 'time'.
There was indeed a drafting error in line 184. The word “time” has been included to give consistency and meaning to the sentence.
- Could the option of the student setting the colour, geometry of the room be a future possibility?
The reviewer raises a very interesting question that invites reflection. Undoubtedly, the research can make it possible for each student to design his or her virtual space. Therefore, it has been included in the conclusion section as an application of this research:
“[..] In this sense, the results may be of interest to design professionals or architects of classrooms in order to maximise the cognitive functions of students, regardless of gender. Research can also contribute to students' ability to shape space in virtual reality classrooms. However, it should not be forgotten that personal preferences of perception do not always coincide with a better level of memory. On the other hand, they may be relevant for teachers looking to adapt their teaching methodology given the existing physical characteristics of the classroom.”
Reviewer 3 Report
The title of the article is quite concise and manages to focus attention on an object of study that, once the article is read in more depth, seems to be the main one or, simply, the one that is addressed to a greater extent throughout the article.
The summary of the article clearly justifies the need or importance of carrying out a study of this nature, and the main objectives, results and most significant conclusions of the study, as a result of reading the summary of the article, are clear and structured.
From the analysis of the theoretical framework section of the article, some conclusions can be deduced that can be quite enlightening and significant:
*The ideas appear organized, structured and contextualized, which causes the reader to have a broad overview of the most significant arguments and information handled in that section of the article.
*The main scientific antecedents of the problem under study, both national and international or regional, are continuously mentioned.
*The current situation or situation in which the problem of the study is contextualized and located is also described with great precision.
*The ideas are frequently supported and endorsed by a wide range of high quality and current bibliographic references, with a periodicity of less than 10 years, in most cases, and complying at all times with the guidelines established and marked in the APA regulations.
*In short, throughout the theoretical framework section of the article, the true objective of the study is made quite clear, thus complying with the most elementary rules of courtesy and good scientific practice.
The objectives of the study are amply described and contextualized throughout the article and in language that is very clear and precise, so that the ambitious aim of focusing the reader very forcefully on the main object of analysis and study of the article is achieved.
The sample that ended up forming part of the study is amply described, and, since it includes a fairly representative number of scientific publications in all the parameters analyzed in the article, it can be considered to have good levels of significance and representativeness, a key element for the final validation of the results derived from the empirical study in question.
The instruments used in the study are extensively described, so that all the instruments that have finally been implemented are quite clear, as well as the objective or purpose for which they were used.
The treatment or analysis of the study data is very well documented and complemented with a wide arsenal of tests and statistical procedures, so that it becomes very clear how the data collected as a result of the implementation of the study were analyzed and treated and, therefore, the information that they managed to report to the study.
The results of the study, as they are written, sequenced and ordered, are somewhat limited and biased towards the main objectives set out in the article. In addition, it is also necessary to highlight that the results described are in line with the most significant methodological approaches established and planned in the study and, therefore, are, for the most part, conveniently organized and written, at least from a narrative, chronological and statistical point of view. Therefore, it is advisable to revise the conclusions section, which is undoubtedly the section with the most shortcomings of the whole.
The discussion, for the most part, is characterized by an extensive critical and broadly contextualized analysis of the most significant results of the study, avoiding, as is required by convention and the most elementary rules of ethics and scientific procedure, a repetitive analysis of the main achievements described in the results section of the article.
The bibliographic references provided in the article are quite extensive and up to date, despite the fact that in some cases they are more than ten years old, and are reflected in the text of the article.
In conclusion, I believe that this empirical article can be published with modifications, especially those related to the conclusions section.
Author Response
We thank the reviewer for your valuable positive comments on the different sections of the article. We appreciate your time and effort to improve the manuscript. In particular, the conclusion section has been updated with these comments and we have also added new paragraphs to improve the document:
“This study analyses the influence of different physical aspects of a classroom on students’ memory from a gender perspective. The results show relevant differences in the effects that design has on memory depending on gender, especially with purple wall colour and low saturation. This provides relevant information for the design of classrooms that benefit both groups.
Lighting, dimensions and colour classroom are fundamental aspects to consider in the design of architectural teaching spaces, but they could also be of interest for the design of virtual learning environments that are increasingly present in our society. The use of VR in this study allows us to explore this. In educational settings, it has been suggested that it may offer even a better experimental context than physical settings [35, 60, 61]. In addition, this has allowed us to do something that was difficult until now: to study the influence of architectural elements on human behavior, overcoming the difficulties and cost of a real construction.
In this sense, the results may be of interest to design professionals or architects of classrooms in order to maximise the cognitive functions of students, regardless of gender.Research can also contribute to students' ability to shape space in virtual reality classrooms. However, it should not be forgotten that personal preferences of perception do not always coincide with a better level of memory. On the other hand, this may be relevant for teachers looking to adapt their teaching methodology given the existing physical characteristics of the classroom.
It would be interesting to analyze the joint effect of the combination of the studied variables and to see if these results are sustained in substantially different classroom typologies. Furthermore, future research may further address this question and whether gender differences are maintained over the years or whether, on the contrary, it is characteristic of youth and limited to learning contexts.”
Reviewer 4 Report
It is noteworthy study to perform physical classroom studies utilizing VR, however, the VR classroom situations aren't appropriate as these wouldn't occur (at least in the United States) in an actual classroom environment. Examples are hue, saturation, and high color temperature, therefore, the conclusions about how this can inform architects and designers is inaccurate. As for the pandemic switching our daily lives to be more at home, it would be noteworthy to document how these design changes can impact a VR classroom if students were to take classes via VR.
Author Response
1. It is noteworthy study to perform physical classroom studies utilizing VR.
We thank the reviewer for your time and appreciation of the importance of the manuscript. We have attended to all the questions and, in addition, the entire expository order of the article has been modified to make it easier to follow.
2. However, the VR classroom situations aren't appropriate as these wouldn't occur (at least in the United States) in an actual classroom environment. Examples are hue, saturation, and high color temperature, therefore, the conclusions about how this can inform architects and designers is inaccurate.
According to the reviewer's comments, we denote difficulty in understanding the impact of this work on architects. Therefore, the conclusions section has been modified to reflect more accurately the relationship between virtual and physical spaces:
“Lighting, colour, and classroom dimensions are fundamental aspects to consider in the design of architectural teaching spaces, but they could also be of interest for the design of virtual learning environments that are increasingly present in our society. The use of VR in this study allows us to explore this. In educational settings, it has been suggested that it may offer even a better experimental context than physical settings [35,60,61]. In addition, this has allowed us to do something that was difficult until now: to study the influence of architectural elements on human behavior, overcoming the difficulties and cost of a real construction.
In this sense, the results may be of interest to design professionals or architects of classrooms in order to maximise the cognitive functions of students, regardless of gender [...]"
3. As for the pandemic switching our daily lives to be more at home, it would be noteworthy to document how these design changes can impact a VR classroom if students were to take classes via VR.
The reviewer raises a very interesting question that invites reflection. The results may be of interest to design the classrooms of the future. The pandemic has further enabled the development of technology for the user. This could also allow students to design customized virtual classrooms according to their preferences. This has been developed in the conclusions section:
“[...] Research can also contribute to students' ability to shape space in virtual reality classrooms. However, it should not be forgotten that personal preferences of perception do not always coincide with a better level of memory. On the other hand, they may be relevant for teachers looking to adapt their teaching methodology given the existing physical characteristics of the classroom”
Round 2
Reviewer 1 Report
Well done!The flow and structure of the paper have been well improved, which helps with readability.